# Stimulator of Interferon Genes Pathway Activation through the Controlled Release of STINGel Mediates Analgesia and Anti-Cancer Effects in Oral Squamous Cell Carcinoma

**DOI:** 10.3390/biomedicines12040920

**Published:** 2024-04-21

**Authors:** Minh Phuong Dong, Neeraja Dharmaraj, Estela Kaminagakura, Jianfei Xue, David G. Leach, Jeffrey D. Hartgerink, Michael Zhang, Hana-Joy Hanks, Yi Ye, Bradley E. Aouizerat, Kyle Vining, Carissa M. Thomas, Sinisa Dovat, Simon Young, Chi T. Viet

**Affiliations:** 1Department of Oral and Maxillofacial Surgery, School of Dentistry, Loma Linda University, Loma Linda, CA 92350, USA; mdong@llu.edu (M.P.D.); mzhang@students.llu.edu (M.Z.); hhanks@students.llu.edu (H.-J.H.); 2Katz Department of Oral Maxillofacial Surgery, The University of Texas Health Science Center at Houston, Houston, TX 77054, USA; neeraja.dharmaraj@uth.tmc.edu (N.D.); jianfei.xue@uth.tmc.edu (J.X.); simon.young@uth.tmc.edu (S.Y.); 3Department of Biosciences and Oral Diagnosis, Institute of Science and Technology, São Paulo State University (Unesp), São Paulo 12245-00, Brazil; estela.tango@unesp.br; 4Department of Chemistry, Rice University, Houston, TX 77005, USA; davidleach@alumni.rice.edu (D.G.L.); jdh@rice.edu (J.D.H.); 5Department of Bioengineering, Rice University, Houston, TX 77005, USA; 6Translational Research Center, Department of Oral and Maxillofacial Surgery, New York University College of Dentistry, New York, NY 10010, USA; yy22@nyu.edu; 7NYU Pain Research Center, Department of Molecular Pathobiology, New York University College of Dentistry, New York, NY 10010, USA; bea4@nyu.edu; 8Department of Preventive and Restorative Sciences, School of Dental Medicine, University of Pennsylvania, Philadelphia, PA 19104, USA; viningk@upenn.edu; 9Department of Materials Science and Engineering, School of Engineering & Applied Science, University of Pennsylvania, Philadelphia, PA 19104, USA; 10Department of Otolaryngology, University of Alabama at Birmingham, Birmingham, AL 35294, USA; carissathomas@uabmc.edu; 11Department of Pediatrics, Pennsylvania State University College of Medicine, Hershey, PA 17033, USA; sdovat@pennstatehealth.psu.edu

**Keywords:** OSCC, STINGel, antinociceptive effect, anti-cancer

## Abstract

Oral squamous cell carcinoma (OSCC) presents significant treatment challenges due to its poor survival and intense pain at the primary cancer site. Cancer pain is debilitating, contributes to diminished quality of life, and causes opioid tolerance. The stimulator of interferon genes (STING) agonism has been investigated as an anti-cancer strategy. We have developed STINGel, an extended-release formulation that prolongs the availability of STING agonists, which has demonstrated an enhanced anti-tumor effect in OSCC compared to STING agonist injection. This study investigates the impact of intra-tumoral STINGel on OSCC-induced pain using two separate OSCC models and nociceptive behavioral assays. Intra-tumoral STINGel significantly reduced mechanical allodynia in the orofacial cancer model and alleviated thermal and mechanical hyperalgesia in the hind paw model. To determine the cellular signaling cascade contributing to the antinociceptive effect, we performed an in-depth analysis of immune cell populations via single-cell RNA-seq. We demonstrated an increase in M1-like macrophages and N1-like neutrophils after STINGel treatment. The identified regulatory pathways controlled immune response activation, myeloid cell differentiation, and cytoplasmic translation. Functional pathway analysis demonstrated the suppression of translation at neuron synapses and the negative regulation of neuron projection development in M2-like macrophages after STINGel treatment. Importantly, STINGel treatment upregulated TGF-β pathway signaling between various cell populations and peripheral nervous system (PNS) macrophages and enhanced TGF-β signaling within the PNS itself. Overall, this study sheds light on the mechanisms underlying STINGel-mediated antinociception and anti-tumorigenic impact.

## 1. Introduction

Oral squamous cell carcinoma (OSCC) is on the rise, increasing by two-thirds in 20 years [1,2], with a substantial increase in young patients who do not have traditional risk factors (i.e., tobacco or alcohol use). OSCC patients suffer from significant morbidity [3] and a 5-year mortality rate of 40% [4], even those at an early stage at diagnosis. Advanced-stage OSCC leads to increased morbidity and mortality due to its potential for local invasion, metastasis, and treatment-related complications. The standard of care for OSCC is surgery followed by risk-adapted radiation and chemotherapy. Few treatment options are available for recurrent or metastatic OSCC that has failed conventional treatment [5]. Immunotherapy with checkpoint inhibitors was approved in 2016 with a modest survival benefit in a subset of patients, thus fueling further research on additional immunotherapeutic strategies, including STING agonism [6].

Aside from poor survival due to treatment resistance, OSCC patients suffer from debilitating pain. Cancer patients who have inadequately managed pain are at risk for opioid tolerance and dependence [7,8,9], anxiety, depression, and reduced quality of life [10,11,12,13,14]. Uncontrolled pain may even contribute to a lower survival rate [15,16]. Cancer patients represent a growing group of people who are prescribed opioids for pain management, of which 21–29% suffer from opioid dependence [17]. Unlike other cancers, OSCC causes intense pain at the primary site at early stages [18], affecting a patient’s ability to eat and speak [8,19]. OSCC patients have higher pain prevalence than other cancer patients [8], require more opioids, and are more likely to suffer from opioid dependence [8,19]. Pain results from either cancer progression [18] or treatment [20]. Previous publications show that the same mechanisms control OSCC carcinogenesis and pain and that OSCC pain is an important early predictor of metastasis and poor survival [21]. Unfortunately, two of the most common drug therapies for OSCC, cisplatin chemotherapy and checkpoint inhibitor anti-PD1 (programmed cell death protein 1) immunotherapy, also exacerbate cancer pain and opioid dependence [22,23]. Cisplatin and opioids have antagonistic actions, with opioids inhibiting the anti-cancer effects of cisplatin in an OSCC model [24]. Anti-PD1 similarly inhibits the mu opioid system by co-localizing PD-1 and mu opioid receptors on small-diameter neurons that convey pain [22]. Anti-PD-1 results in opioid-induced hyperalgesia and opioid tolerance in mice and non-human primates [23]. Pain at the cost of cancer treatment is detrimental to patients [7]. The optimal therapeutic agent possesses the capability of targeting both oncogenic pathways and pain-modulating pathways, leading to both tumor growth inhibition and cancer pain relief.

The stimulator of interferon genes (STING) plays a crucial role in the innate immune response and facilitates the activation of immune cells and the production of cytokines, emerging as a candidate for immunotherapy [25,26]. In addition to its anti-tumor properties, STING agonism has the ability to reduce bone cancer pain via immune and neuronal modulation in an in vivo model [27]; however, the cell compositions and functional pathways involved are not well characterized. Biomaterials have been engineered to deliver pain relief in many diseases [28,29]. Multidomain peptides (MDPs) are a class of self-assembling peptide hydrogels that have been well characterized [30,31]. STINGel is a novel biomaterial-based drug delivery system that utilizes the MDP hydrogel-based platform for controlled-release delivery of the STING agonist CDN (Cyclic DiNucleotide dithio-(RP,RP)-[cyclic[A(2′,5′)pA(3′,5′)p) [6]. Previous studies demonstrate that STINGel has a robust anti-tumor effect in a syngeneic HPV-associated OSCC mouse model [6]. The role of the STINGel in mediating OSCC pain, however, has not been characterized. Clinical translation of STINGel requires an understanding of how it modulates cancer symptoms.

In this study, we hypothesized that STINGel treatment, owing to its extended-release formulation, produces a sustained antinociceptive and anti-tumor effect. We tested the effect of STINGel in two OSCC mouse models. We then used single-cell RNA sequencing analysis to determine the immune cell panel and signaling pathways that were responsible for the antinociceptive and anti-tumor activity of STINGel treatment.

## 2. Materials and Methods

### 2.1. STINGel Preparation

STINGel was prepared as previously described [6]. Briefly, the peptide portion with sequence KKSLSLSLSLSLSLKK was synthesized by standard solid-phase peptide synthesis using an FMOC protection strategy on Rink Amide MBHA resin and characterized by HPLC and MALDI-TOF MS. A buffered mixture of this peptide and CDN were made such that the final concentrations were 10 μg/μL peptide, 0.67 μg/μL CDN, 0.5× HBSS (Hank’s balanced salt solution), and 149 mM sucrose. This mixture rapidly forms the gel composition used in this study. The 30 μL boluses used in this study thus provide 20 μg of CDN.

### 2.2. Cell Culture

MOC1, the murine oral cancer cell line, was obtained from Dr. Ravindra Uppaluri (Dana-Farber Cancer Institute, Harvard University, Boston, MA, USA). Cells were grown in IMDM/F12 (2:1) (HyClone, Logan, UT, USA) with 5% fetal bovine serum (HyClone), penicillin/streptomycin (Lonza, Walkersville, MD, USA), 5 ng/mL EGF, 400 ng/mL hydrocortisone (Sigma, St. Louis, MO, USA), and 5 µg/mL insulin [32] at 37 °C in 5% CO_2_. Cells at 80–95% confluency were used for all experiments.

### 2.3. Mouse Cancer Model

All animal protocols and experimental procedures were approved by the Loma Linda University Institutional Animal Care and Use Committee (IACUC), in accordance with the Guide for the Care and Use of Laboratory Animals. Mice were housed in a temperature-controlled environment, alternating 12 h light–dark cycle with free access to water and a standard rodent diet.

Mouse cancer models were performed in female mice (C57BL/6 strain) at 6–8 weeks of age. Mice were injected with MOC1 tumor cells on day 0 into the right maxillary oral vestibule [33] or right hind paw (30,000 cells in 30 µL volume), followed by CDN/STINGel (20 μg CDN in 30 µL injection) or HBSS/control on post-inoculation day (PID) 3 (when the tumors were 4–5 mm in size) in the same inoculated location. The nociceptive behavioral assays were performed on day 0 (prior to inoculation), day 4, and then every 3 days after STINGel or vehicle injection. The mice were sacrificed on day 30 and tissues were harvested.

### 2.4. Nociceptive Behavioral Assays

#### 2.4.1. Thermal Nociception Assay

Mice were placed in a plastic chamber on a 25 °C glass surface. Thermal hyperalgesia was measured by the paw thermal stimulator (IITC Life Sciences, Woodland Hills, CA, USA) with a radiant heat source delivering a thermal stimulus to the left hind paw of each mouse with a cutoff of 20 s [34]. Paw withdrawal latency was measured as a mean of 3 trials taken at 5 min intervals.

#### 2.4.2. Mechanical Nociception Assay

Mice were acclimated in a plastic cage with a wire mesh floor for 1 h. Paw withdrawal thresholds [35] were determined in response to pressure from von Frey filaments (IITC Life Sciences). The paw withdrawal threshold for each mouse was determined as the mean of 3 trials.

#### 2.4.3. Facial Mechanical Nociception Assay

Over a period of two weeks, mice were acclimated every other day in a transparent box with a mesh floor for 1 h. To assess their withdrawal responses to mechanical stimulation, von Frey filaments ranging from 0.0008 to 4 g-force (totaling 11 filaments) were applied in ascending order to the cheek area [36]. Each von Frey filament was applied once; in case of a moving mouse or unclear response, the same filament was reapplied to the same area after the initial stimulus g. Different intensities were set at 5 min intervals. The facial nociception score was reported as a numerical average of the 11 responses ranging from 0 (no response) to 4 (multiple facial grooming, responding to the filament simulation with more than three facial wipes continuously) [34].

### 2.5. Single-Cell Suppression and Droplet-Based Single-Cell RNAseq

A MOC1 orthotopic tumor model was established, and treatments were performed as described earlier [33]. Cells were isolated from mouse oral tumors, one normal HBSS control and one treated with STINGel biomaterial. The tumors were harvested and then processed into a single-cell suspension, and red blood cells were lysed, followed by a bead-based enrichment for CD45^+^ leukocytes. Cells were resuspended to a concentration of approximately 1000 cells/µL and loaded onto a 10× Genomics Chromium platform for droplet-enabled scRNA-Seq. To perform single-cell analysis, cell capture, lysis, reverse transcription, and cDNA amplification were performed at the Single Cell Genomics Core (SCGC) at the Baylor College of Medicine (BCM). Library generation was performed at the Genomic and RNA Profiling Core (GARP) at the BCM. A total of 2689 cells were profiled for the control and 4336 cells were profiled for the treatment mouse.

### 2.6. Single-Cell RNA Sequence Data Analysis

The CellRanger (v.3.0.2; 10× Genomics Inc., Pleasanton, CA, USA) pipeline was used to process the data. Initially, samples in each pool were demultiplexed using the sample index, and then a count matrix was generated for each sample by mapping to the mm10 reference genome. The filtered expression matrices were loaded into R v4.3.1 using the ‘Read10X’ function from the Seurat v4.3.0.1 package [37] with pooled data count from 2 batches (7234 cells). 

Cells with less than 200 non-zero genes or more than 6% of mitochondrial genes were filtered out; cells with unique UMI counts under 500 or greater than 10,000 were also filtered out. Genes expressed in less than 10 cells were omitted. The doublets were detected by DoubletFinder v2.0.3 [38]. After filtering, the data contained 6796 cells and 17,297 genes. Data were then log normalized and scaled to regress out cell cycle, percentage mitochondria, and number of features (genes). Variable genes were identified using the FindVariableFeatures function. The Uniform Manifold Approximation and Projection (UMAP) dimensional reduction and graph-based clustering method (resolution = 1.25) were computed using the top 24 principal components. The FindAllMarkers function was used to calculate the marker genes (thresh.use > 0.25, min.pct > 0.25, Wilcoxon rank-sum test). Finally, we annotated each cell type by SingleR v2.0.0 using the ImmuGen database [39], with an extensive literature search for the specific gene expression patterns.

Cell communication analysis was performed using the R package Cellchat v1.6.1 [40] with the mouse CellChatDB.

Enrichment analysis was performed using clusterProfiler package v4.8.2 [41].

### 2.7. Statistical Analysis

Statistical analysis was performed using GraphPad Prism v9.5.0. The thermal and mechanical nociception scores of the following days were converted into percentage changes compared to day 0. The differences between the thermal and mechanical scores of the STINGel and vehicle groups were analyzed using two-way ANOVA and Tukey’s post hoc test. Results were presented as mean ± standard error of the mean (SEM). A *p*-value of lower than 0.05 was considered to be statistically significant.

## 3. Results

### 3.1. STINGel Treatment Reduced Facial Mechanical Allodynia in an Orofacial OSCC Mouse Model

The maxillary vestibule OSCC model was used to assess the antinociceptive and anti-tumor effect of STINGel in the orofacial region, an anatomically synonymous model of orofacial pain in oral cancer patients. Mechanical allodynia was quantified using the facial mechanical withdrawal assay. The percentage change in facial mechanical withdrawal score was calculated based on the values of day 0 prior to cancer inoculation with MOC1 cells, with increasing percentage changes signifying increased pain. The results showed that by PID 4 when there were visible tumors, the mice in both groups had increased facial nociception, which gradually increased throughout the experimental time course. However, the STINGel treatment group had significantly reduced facial allodynia with the nociceptive effect at only 50% compared with the vehicle hydrogel (MDP) group (*p* < 0.0001) (Figure 1A). Moreover, the treatment group displayed a statistically significant reduction in tumor volume compared to the vehicle group on day 40 (Appendix A). This anti-tumor effect of STINGel in OSCC preclinical models has been previously reported [33].

### 3.2. STINGel Treatment Reduced Thermal Hyperalgesia and Mechanical Allodynia in an Orthotopic OSCC Mouse Model

We determined the antinociceptive effect of STINGel in an orthotopic paw cancer model. Thermal hyperalgesia and mechanical allodynia were quantified by the thermal and paw withdrawal assay, respectively, in mice inoculated with MOC1 cells into the hind paw. On post-inoculation day (PID) 4, mice had remarkably increased nociception, with reduced thermal latency and mechanical thresholds at approximately 50% and 80% from the baseline, respectively. At this point, the mice were injected with STINGel or vehicle hydrogel (MDP) (Figure 1B,C). On PID 7 (day 3 of STINGel treatment), the STINGel group showed significantly less thermal hyperalgesia and mechanical allodynia in comparison to the vehicle group. Notably, the thermal latency in the STINGel group increased back to the pre-inoculation baseline, while the thermal thresholds of the vehicle group were still at 40% below day 0. In the following days, the thermal and mechanical withdrawal thresholds of the STINGel group remained stable with a sustained antinociceptive effect. Overall, the thermal and mechanical withdrawal thresholds of the vehicle group were significantly lower than that of the STINGel group, indicating worse nociception. 

### 3.3. STINGel Treatment Changed the Ratio of M1/M2 Macrophages and the Population of N1-like Neutrophils in the Mouse OSCC Model

The maxillary vestibule OSCC model was used to further assess the mechanism of STINGel-induced anti-tumor and antinociceptive effects. CDNs have been known to activate immune cells in preclinical models through STING. While the mechanism of STING agonist-mediated anti-tumor effect is well characterized, the effect of controlled release formulation STINGel on immune cell infiltrate is not yet defined. Furthermore, the cellular mechanism of STING agonist-mediated antinociception in cancer is not well defined. Using the Seurat package, the single-cell RNA-seq data were normalized, pooled, and clustered (Appendix A). In order to broadly annotate these populations, canonical markers were employed, resulting in the annotation of T cells (Cd3d, Cd3e), NK cells (Gzma, Nkg7), dendritic cells (Wdfy4, Flt3), macrophages/monocytes (Lyz2, Csf1r), neutrophils (Csf1, Cfs3r), and a cluster expressing Siglech and Tgfbr, which are considered PNS macrophages (Figure 2A and Appendix A). We applied SingleR using the ImmGen database as the reference to confirm the annotation of cell clusters (Figure 2B) and generated assignment scores across all cell label combinations (Figure 2C).

Macrophages/monocytes were categorized into distinct subtypes, including M1-like (Cd68, Ctsl, Pf4), M2-like (Arg1, Ccl24), Retnla+ macrophages (Retnla), and monocytes (Ly6c2) (Figure 3A,B and Appendix A). An observation emerged from the comparison between STINGel-treated and untreated samples (Figure 3C), revealing a significant increase in total monocyte population and an elevated M1-like/M2-like macrophage ratio in the STINGel-treated mouse (vehicle: 0.16, STINGel: 3.35). Additionally, STINGel treatment tissues were enriched by 5.5 folds in proportions of N1-like neutrophils (Tnf, Ccl3) compared to vehicle tissues, while N2-like neutrophils proportions remained unchanged (Cxcr4, Mmp9) [42,43].

### 3.4. Regulatory Pathways Involved in STINGel Treatment

To further investigate the biological functions of differential gene expression after STINGel treatment (adjusted *p* < 0.05, log foldchange > 0.1 as the cut-off criterion), we performed over-representation analysis (ORA) and Gene Set Enrichment Analysis (GSEA) using Gene Ontology (GO) in macrophages, monocytes, and neutrophils, which have roles in modulating pain by producing pro- or anti-inflammatory mediators to promote or resolve pain [44]. 

The differentially expressed genes in M1-like macrophages and monocytes after STINGel treatment were responsible for the activation of the immune response, myeloid cell differentiation, and cytoplasmic translation (Figure 4 and Appendix A). Additionally, based on over-representation analysis, STINGel treatment induced significant modifications in leukocyte migration, the cytokine-mediated signaling pathway, and leukocyte-mediated immunity pathway activities in N1 neutrophils (Appendix A). STINGel treatment also led to discernible changes in the translation regulation of the pre-synapse/synapse/post-synapse pathway within M1 macrophages, monocytes, and N2 neutrophils (Figure 4, Appendix A).

Next, we investigated the GSEA GO in M2-like macrophages, a subset of immune cells with immunosuppressive properties, releasing anti-inflammatory cytokines and growth factors to facilitate tissue repair and alleviate pain [44]. We identified the top 20 enriched pathways, revealing that STINGel treatment led to the suppression of translation at synapses while activating the pathway involved in the negative regulation of neuron projection development (Figure 5). A similar suppression pattern of translation at synapses was also observed in PNS macrophages and Retnla+ macrophages (Appendix A). These results indicate that STINGel-mediated antinociception was associated with the suppression of translation at synapses and regulation of neuron projection development. 

### 3.5. Cell–Cell Communication

We applied CellChat to investigate the cell–cell communication in STINGel treatment compared to vehicle treatment. The analysis revealed that the number and strength of interactions increased in STINGel treatment (Appendix A). Under the influence of STINGel, M1-like macrophages and N1-like neutrophils, two cell populations that are believed to be increased in the setting of tumor killing, exhibit heightened interaction potency, leading to a significant increase in the strength of both incoming and outgoing signals (Figure 6A). The incoming and outgoing signals included CCL, Secreted Phosphoprotein 1 (SPP1), Macrophage Migration Inhibitory Factor (MIF), and TNF pathway signals (Figure 6B and Appendix A). Remarkably, STINGel treatment induced a significant upregulation of TGF-β pathway signaling from M1-like cells, M2-like cells, and monocytes to PNS macrophages, accompanied by a concomitant enhancement of TGF-β signaling within the PNS macrophages themselves. These enhanced signals were absent in the vehicle treatment group (Figure 6C).

## 4. Discussion

STING plays a crucial role in the detection of cytosolic DNA and the activation of the innate immune response [45]; preclinical studies [6,33,46] have demonstrated that DMXAA or ADU-S100 (STING agonists) have a significant anti-tumor effect [26,27,47]. Our study also showed a significant reduction in tumor volume in mice treated with STINGel, which recapitulates findings from our previous studies in multiple preclinical models of oral SCC. STING recruits neutrophils, followed by monocytes, CD8 T cells [47], and M1-like cells to the tumor in preclinical models [48,49]. M1 macrophages and N1 neutrophils are subsets of immune cells that produce robust pro-inflammatory responses and enhance immune responses against pathogens, contributing to the anti-tumor effect [50]. On the other hand, M2 macrophages can create an immunosuppressive environment within the tumor, limiting the effectiveness of an anti-tumor immune response [50]. The utilization of STING agonists in murine cancer models requires repetitive administration through multiple injections, resulting in limited effects on tumor shrinkage and survival [6]. STINGel, a multidomain peptide hydrogel loaded with cyclic dinucleotides (CDNs), facilitates the controlled release of CDN delivery and has exhibited a six-fold enhanced overall survival rate in a murine oral cancer model when compared to CDN monotherapy [33]. In this study, the M1 macrophages and N1 neutrophil populations were increased after STINGel treatment (Figure 3C). Additionally, GO pathway analysis revealed that gene changes in M1 macrophages and N1 neutrophils after STINGel treatment were related to activation of immune response, myeloid cell differentiation, and leukocyte migration (Figure 4 and Appendix A). Cell–cell communication analysis showed that M1-like phenotypes and N1-like phenotypes upregulated SPP1, MIP, and TNF signaling pathways (Figure 6B and Appendix A). Although characterized as pro-nociceptive pain inducers due to their ability to secrete cytokines, M1 and N1 cells in the tumor microenvironment have the potential to enhance anti-tumor immunity and minimize tumor burden. These non-neuronal, immunomodulatory effects may inhibit pain by reducing the production of cancer cell-derived pain mediators. Altogether, STINGel demonstrates its indirect antinociceptive impact by reducing the tumor burden through enhancing the presence of M1-like macrophages and N1-like neutrophils and concurrently activating immune response pathways through the upregulation of TNF and MIF signaling pathways.

A previous publication has shown that mice deficient in STING signaling had increased sensitivity to nociceptive stimuli and heightened excitability to nociceptors; conversely, the intrathecal activation of STING led to significant an antinociceptive effect in both mice and non-human primates [51]. In a preclinical bone cancer model, STING agonists attenuate acute pain by directly affecting neuronal modulation and provide long-term cancer pain relief through influencing the immune cell function [27]. To extend these initial findings of the antinociceptive properties of STING agonists, our study used STINGel to test the effect of controlled-release STING agonists focused on a soft tissue cancer pain model. STINGel potently reduced mechanical and thermal hyperalgesia in our oral cancer models after 3 days of treatment, and this antinociceptive effect was stable until day 30 (Figure 1). Additionally, GO analysis demonstrated that STINGel affected translation at pre-synapse, synapse, and post-synapse in macrophages, monocytes, and N2 neutrophils. GSEA in M2-like macrophages demonstrated that STINGel regulated neuron projection development and regeneration (Figure 5). A significant proportion of immune cells, such as macrophages and neutrophils, are determined to express receptors for neurotransmitters and neuropeptides on their cell surface [52]. Recent investigations indicated that neurotransmitters, miRNAs, and neuropeptides from nociceptors have the capacity to modulate immune responses [53]. Further, another publication demonstrated that they contribute to the facilitation of cancer progression via the suppression of immune functions [54,55,56]. STINGel treatment shows its function in regulating the neuro-immune axis.

TGF-β is known as an anti-inflammatory cytokine that reduces the synthesis of pro-inflammatory cytokines [57]. Significant evidence substantiates TGF-β1’s relevance as a mediator of nociceptive processes and its potential as a deterrent against the development of chronic neuropathic pain. Its mechanisms include the attenuation of neuroimmune responses in both neurons and glia, coupled with the facilitation of endogenous opioid expression within the spinal cord [58,59,60]. The secretion of TGF-β from bone marrow stromal cells exerts neuromodulatory effects, leading to the inhibition of dorsal root ganglion hyperexcitability through noncanonical signaling mediated by TGF-β receptor 1 and the attenuation of neuropathic pain [61]. In this study, TGF-β was upregulated after STINGel treatment; monocytes and M1-like and M2-like macrophages signaled to PNS macrophages through TGF-β (Figure 6C) [62]. We demonstrated that the upregulation of the TGF-β signaling pathway in PNS macrophages contributes to the antinociceptive effect of STINGel. The current study concentrated on a solitary time point following STINGel treatment. There is a need for future comprehensive studies to delve into STINGel effects during the entire course of cancer treatment.

This study investigated the impact of STINGel on cancer pain in a maxillary vestibule OSCC model. By examining the changes in immune cell populations and regulatory pathways, this study identifies significant mechanisms underlying the antinociceptive and anti-tumor effects of STINGel treatment. These results will pave the way for the development of new therapeutic strategies for cancer pain management and cancer treatment in OSCC patients. However, this study has some limitations. The findings are based on animal models and primarily focuse on a single time point after STINGel treatment, which may not fully represent the complexities of the STINGel effect following treatment.

## 5. Conclusions

In conclusion, this study provides insights into the impact of STINGel treatment on cancer pain in a soft tissue cancer model. STINGel exhibits anti-tumor properties by elevating the population of M1-like macrophages and N1-like macrophages and stimulating the immune response pathway. STINGel-mediated antinociception was likely mediated by the regulation of neuron development and TGF-β signaling to PNS macrophages.

## Figures and Tables

**Figure 1 biomedicines-12-00920-f001:**
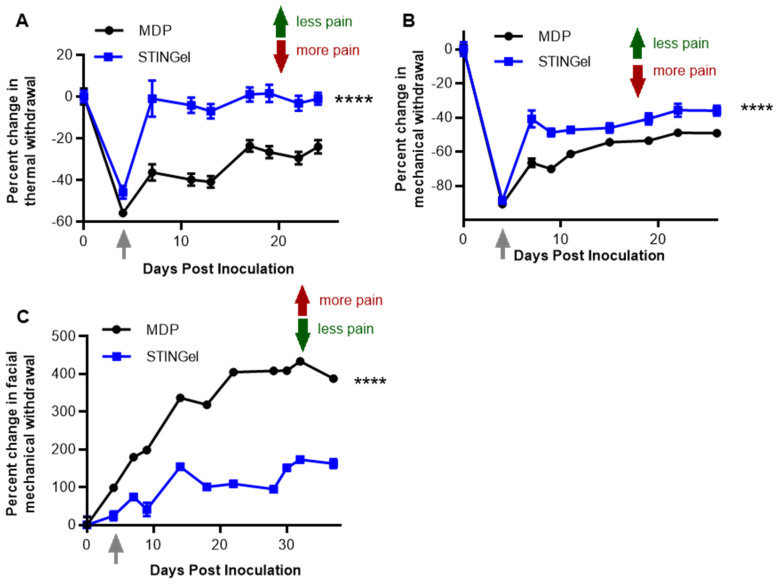
STINGel treatment mitigated the pain in mouse oral squamous cell cancer models. Effects of STINGel (blue line) in comparison to vehicle (black line) on (**A**) facial mechanical nociception (maxillary vestibule model), (**B**) thermal nociception (paw hind model), and (**C**) paw withdrawal (paw hind model). The arrow indicates the time of STINGel or MDP (vehicle) injection. The dots show the mean values; the error bars indicate the standard error of the mean. *n* = 7–8 per group. Two-way ANOVA and Tukey’s multiple comparisons were used, **** *p* < 0.0001.

**Figure 2 biomedicines-12-00920-f002:**
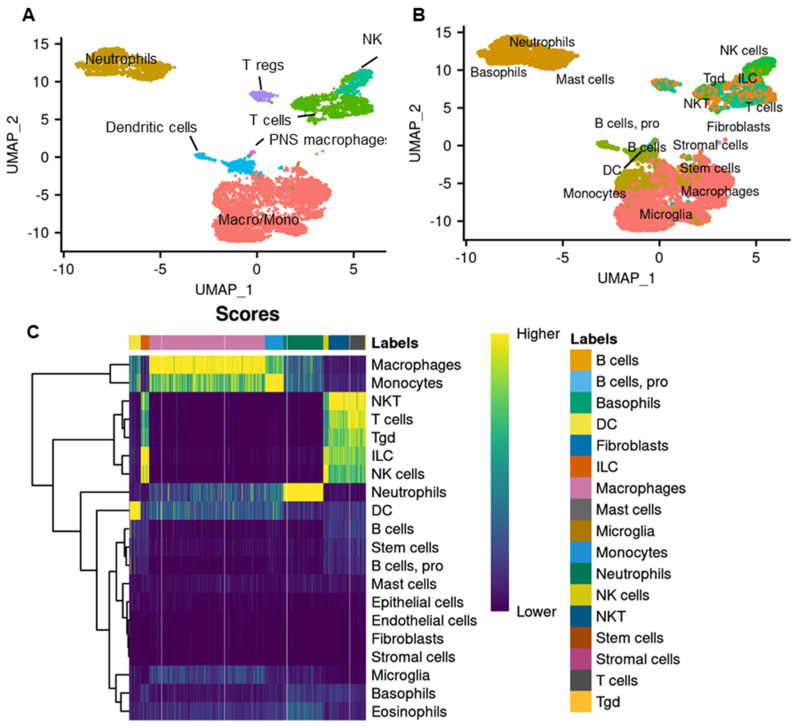
Single-cell RNA sequencing analysis of STINGel treatment and vehicle in a mouse oral squamous cell cancer model. (**A**) Uniform manifold approximation and projection (UMAP) plot with clusters denoted by colors and labeled according to canonical markers. (**B**) UMAP plot with SingleR annotations indicated for individual cells. (**C**) Heatmap of SingleR scores for the top correlates cell types; each cell is a column, while each row is a label in the reference of the ImmuGen dataset, and the final label for each cell is shown in the top bar.

**Figure 3 biomedicines-12-00920-f003:**
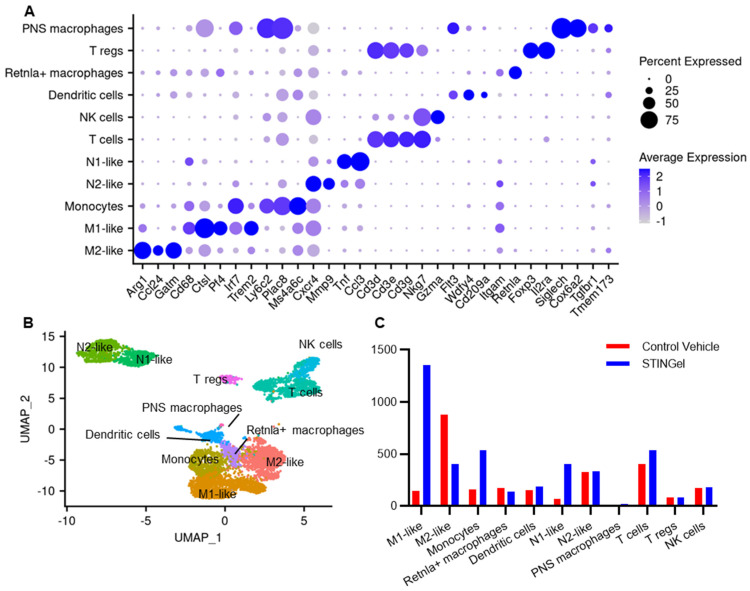
STINGel treatment changed the population of monocytes, macrophages, and neutrophils in the mouse oral squamous cell cancer model. (**A**) Dot plot of average expression of the canonical markers of each cell type. The relative gene expression in percent is represented by the size of dots. The average expression level is indicated by the color. (**B**) Uniform manifold approximation and projection (UMAP) plot with 11 cell types denoted by color and labeled according to canonical markers. (**C**) The bar plot represents the number of each cell type in the condition of STNGel treatment (blue) or vehicle (red).

**Figure 4 biomedicines-12-00920-f004:**
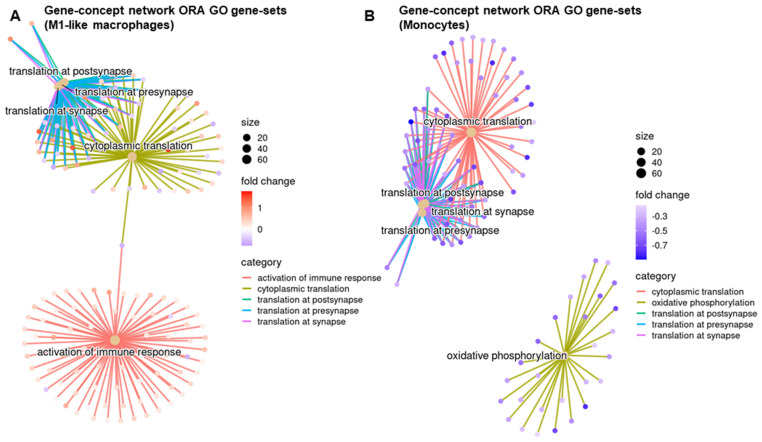
Enrichment analysis of Gene Ontology (GO) in M1-like macrophages and monocytes. Differentially expressed genes based on the treatment of (**A**) M1-like macrophages and (**B**) monocytes underwent the gene concept network of over-representation analysis of GO biological process pathways.

**Figure 5 biomedicines-12-00920-f005:**
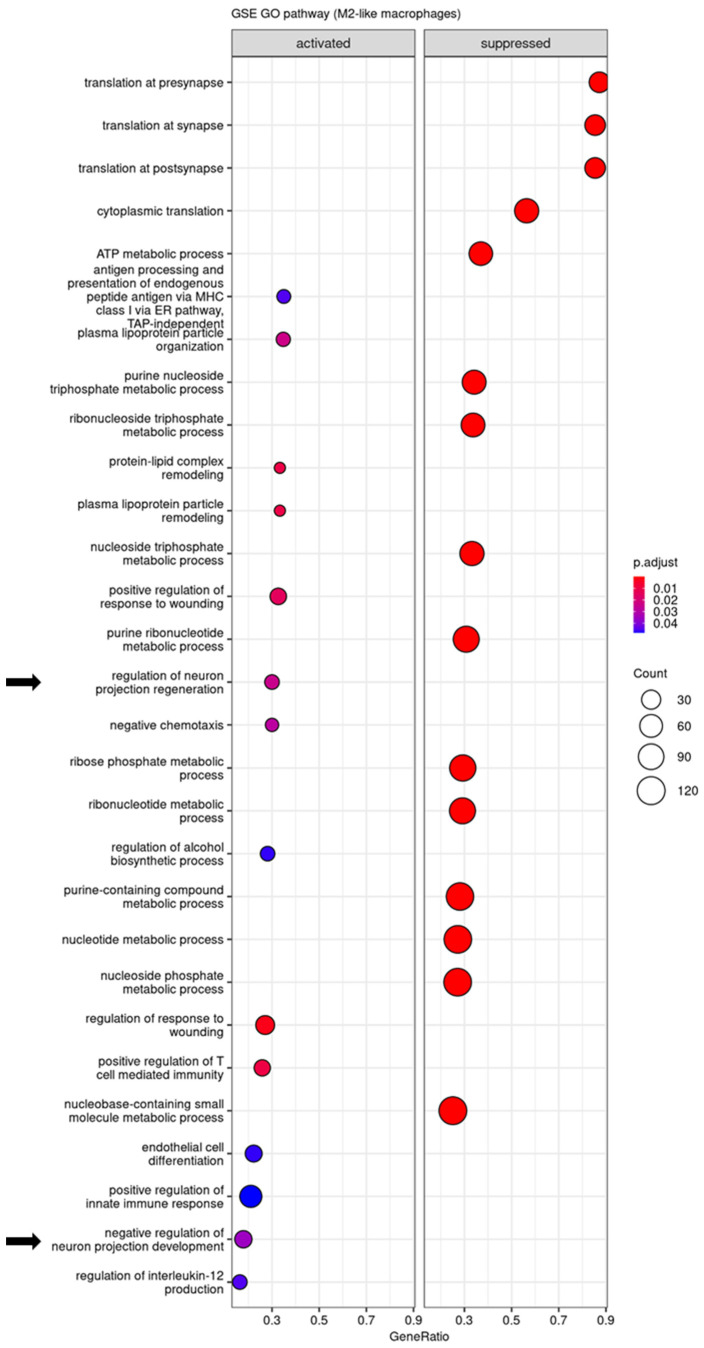
Enrichment analysis of Gene Ontology (GO) in M2-like macrophages. Dot plot of gene set enrichment GO biological process pathways in M2-like macrophages reveals that STINGel treatment is associated with the regulation of neuron projection development and regeneration (arrow).

**Figure 6 biomedicines-12-00920-f006:**
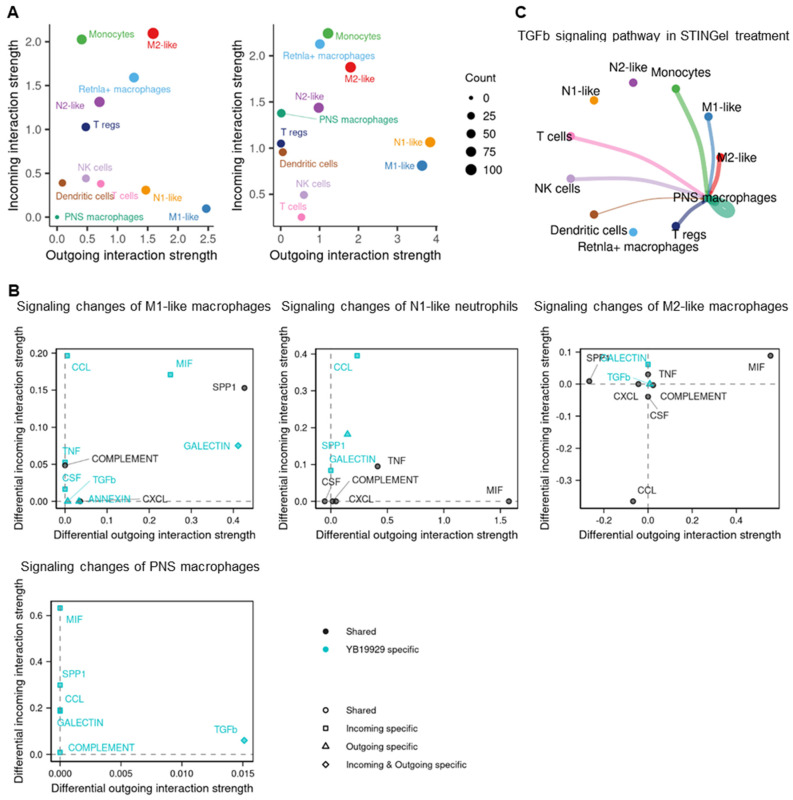
The difference in cell-cell interaction between STINGel treatment and vehicle. (**A**) Scatter plot of the outgoing and incoming interaction strength of cell clusters in the mouse oral squamous cell carcinoma model with and without STINGel treatment. (**B**) Scatter plots of specific signaling changes of M1-like macrophages (upper left panel), N1-like neutrophils (upper middle panel), M2-like macrophages (upper right panel), and PNS macrophages (lower left panel) between STINGel treatment and vehicle. (**C**) Circle plot of the significant ligand-receptor pairs for TGF-β between each cell type in STINGel treatment. Edge colors are consistent with the sources as the sender, and edge weights are proportional to the interaction strength. A thicker edge line indicates a stronger signal.

## Data Availability

All data generated for this study are included in this article. The raw data supporting the conclusions of this article will be made available by the authors on request.

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
