# Peer review of "Stimulator of Interferon Genes Pathway Activation through the Controlled Release of STINGel Mediates Analgesia and Anti-Cancer Effects in Oral Squamous Cell Carcinoma"

_biomedicines, 2024, doi:10.3390/biomedicines12040920_

Round 1

Reviewer 1 Report

Comments and Suggestions for Authors

I commend the authors on their work. The research question is clear and the methodology is sound and well documented. The manuscript reads well, and I'm happy to support its publication. However, the authors must revise the text since they had a high similarity rate. For example, the Facial mechanical nociception assay section is almost copied and pasted from another article. 

Author Response

1. Summary

2. Questions for General Evaluation

Reviewer’s Evaluation

Response and Revisions

Does the introduction provide sufficient background and include all relevant references?

Yes

Are all the cited references relevant to the research?

Yes

Is the research design appropriate?

Yes

Are the methods adequately described?

Yes

Are the results clearly presented?

Yes

Are the conclusions supported by the results?

Yes

3. Point-by-point response to Comments and Suggestions for Authors

Comments 1: I commend the authors on their work. The research question is clear and the methodology is sound and well documented. The manuscript reads well, and I'm happy to support its publication. However, the authors must revise the text since they had a high similarity rate. For example, the Facial mechanical nociception assay section is almost copied and pasted from another article. 

Response 1: Thank you for pointing this out. We agree with this comment. We used the same methods from a previously published manuscript by our own group, so the wording is highly similar. We have changed the facial mechanical nociception assay section on pages 3-4, lines 141-149 in the revised manuscript, as below.

Over a period of two weeks, mice were acclimated every other day in a transparent box with a mesh-floor for 1 hour. To assess their withdrawal responses to mechanical stimulation, von Frey filaments ranging from 0.0008 to 4 g-force (totaling 11 filaments) were applied in ascending order to the cheek area [36]. Each von Frey filament was applied once; in case of moving mouse or unclear response, the same filament was reapplied to the same area after the initial stimulus g. Different intensities were set at 5 min intervals. The facial nociception score was reported as a numerical average of the 11 responses ranging from 0 (no response) to 4 (multiple facial grooming, responding to the filament simulation with more than three facial wipes continuously) [34].”

Reviewer 2 Report

Comments and Suggestions for Authors

This study investigates the impact of intra-tumoral STINGel on OSCC-induced pain using two separate OSCC models and nociceptive

behavioral assays.

The paper is neatly written.

According to iThenticate, this manuscript is repeated in relation to other works by 40%, which is too much, please correct it?

Please describe the statistical processing in more detail.

Fig 5 is illegible. Please increase the font size on all figures.

What are the limitations and strengths of this study.

The literature is listed incorrectly.

Author Response

1. Summary

Thank you very much for taking the time to review this manuscript. Please find the detailed responses below and the corresponding revisions/corrections highlighted/in track changes in the re-submitted file.

2. Questions for General Evaluation

Reviewer’s Evaluation

Response and Revisions

Does the introduction provide sufficient background and include all relevant references?

Can be improved

Are all the cited references relevant to the research?

Can be improved

Is the research design appropriate?

Can be improved

Are the methods adequately described?

Can be improved

Are the results clearly presented?

Must be improved

Are the conclusions supported by the results?

Can be improved

3. Point-by-point response to Comments and Suggestions for Authors

Comments 1: This study investigates the impact of intra-tumoral STINGel on OSCC-induced pain using two separate OSCC models and nociceptive behavioral assays. The paper is neatly written. According to iThenticate, this manuscript is repeated in relation to other works by 40%, which is too much, please correct it?

Response 1: We have revised the method section on pages 3-4, lines 141-149, 168-173, and the introduction section on page 2, lines 80-81 in the revised manuscript, as below.

Lines 80-81 “The stimulator of interferon genes (STING) plays a crucial role in the innate immune response and facilitates the activation of immune cells and the production of cytokines, emerging as a candidate for immunotherapy [25, 26]”

Lines 141-149 “Over a period of two weeks, mice were acclimated every other day in a transparent box with a mesh-floor for 1 hour. To assess their withdrawal responses to mechanical stimulation, von Frey filaments ranging from 0.0008 to 4 g-force (totaling 11 filaments) were applied in ascending order to the cheek area [36]. Each von Frey filament was applied once; in case of moving mouse or unclear response, the same filament was reapplied to the same area after the initial stimulus g. Different intensities were set at 5 min intervals. The facial nociception score was reported as a numerical average of the 11 responses ranging from 0 (no response) to 4 (multiple facial grooming, responding to the filament simulation with more than three facial wipes continuously) [34]”

Lines 168-173 “Cells with less than 200 non-zero genes or more than 6% of mitochondrial genes were filtered out; cells with the unique UMI counts under 500 or greater than 10000 were also filtered out. Genes expressed in less than 10 cells were omitted. The doublets were detected by DoubletFinder v2.0.3 [38]. After filtering, the data contained 6796 cells and 17297 genes. Data were then log-normalized and scaled to regress out cell cycle, percentage mitochondria, and number of features (genes).”

Comments 2: Please describe the statistical processing in more detail.

Response 2: We have revised the statistical analysis section on page 4, lines 184-189 in the revised manuscript, as below.

Statistical analysis was performed using GraphPad Prism v9.5.0. The thermal and mechanical nociception scores of the following days were converted into percentage changes compared to day 0. The differences between the thermal and mechanical scores of the STINGel and vehicle groups were analyzed using two-way ANOVA and Tukey’s post hoc test. Results were presented as mean ± standard error of the mean (SEM). A p-value of lower than 0.05 was considered to be statistically significant.”

Comments 3: Fig 5 is illegible. Please increase the font size on all figures.

Response 3: We have increased the font size of all figures in the revised manuscript.

Comments 4: What are the limitations and strengths of this study.

Response 4: We have revised and stated the strengths and limitations of our study in the Discussion section, page 12, lines 393-400 in the revised manuscript, as below

“This study investigated the impact of STINGel on cancer pain in a maxillary vestibule OSCC model. By examining the changes in immune cell populations and regulatory pathways, the study identifies significant mechanisms underlying the antinociceptive and antitumor effects of STINGel treatment. These results will pave the way for the development of new therapeutic strategies for cancer pain management and cancer treatment in OSCC patients. However, this study has some limitations. The finding is based on animal models and primarily focuses on a single time point after STINGel treatment, which may not fully represent the complexities of STINGel effect following treatment.”

Comments 5: The literature is listed incorrectly.

Response 5: We have changed the reference format according to the journal's requirements in the revised manuscript.

Round 2

Reviewer 2 Report

Comments and Suggestions for Authors

Thanks to the authors for the corrections. Congratulations on the manuscript. I ask the editor to pay attention to the large percentage of coincidences before publication (40%). If this is not a problem to post, ignore my comment.